# The Controversial Role of LPS in Platelet Activation In Vitro

**DOI:** 10.3390/ijms231810900

**Published:** 2022-09-17

**Authors:** Luca Galgano, Gianni Francesco Guidetti, Mauro Torti, Ilaria Canobbio

**Affiliations:** 1University School for Advanced Studies IUSS, 27100 Pavia, Italy; 2Department of Biology and Biotechnology “Lazzaro Spallanzani”, University of Pavia, 27100 Pavia, Italy

**Keywords:** platelets, LPS, TLR4, *E. coli*, thromboinflammation, sepsis

## Abstract

Circulating platelets are responsible for hemostasis and thrombosis but are also primary sensors of pathogens and are involved in innate immunity, inflammation, and sepsis. Sepsis is commonly caused by an exaggerated immune response to bacterial, viral, and fungal infections, and leads to severe thrombotic complications. Among others, the endotoxin lipopolysaccharide (LPS) found in the outer membrane of Gram-negative bacteria is the most common trigger of sepsis. Since the discovery of the expression of the LPS receptor TLR4 in platelets, several studies have investigated the ability of LPS to induce platelet activation and to contribute to a prothrombotic phenotype, per se or in combination with plasma proteins and platelet agonists. This issue, however, is still controversial, as different sources, purity, and concentrations of LPS, different platelet-purification protocols, and different methods of analysis have been used in the past two decades, giving contradictory results. This review summarizes and critically analyzes past and recent publications about LPS-induced platelet activation in vitro. A methodological section illustrates the principal platelet preparation protocols and significant differences. The ability of various sources of LPS to elicit platelet activation in terms of aggregation, granule secretion, cytokine release, ROS production, and interaction with leukocytes and NET formation is discussed.

## 1. Introduction

Platelets are small anucleated blood cells that circulate in the bloodstream at concentrations of 150,000–400,000/μL and are mainly responsible for hemostasis and thrombosis [1,2,3]. Platelets are also primary sensors of invading agents, such as bacteria, viruses, and fungi, and are critically involved in inflammation, innate immunity [4,5], and sepsis [6,7,8]. Sepsis is a complex condition of organ dysfunction caused by pathological and biochemical impairments consequent to infection, and it is often associated with severe thrombotic complications [7,9]. Bacteria are the most common pathogens causing sepsis, and *Staphylococcus aureus*, *Streptococcus* spp., and *Escherichia coli* are the most frequent infective agents. *E. coli* induces sepsis through the release of the bacterial endotoxin lipopolysaccharide (LPS), a common component of the outer membrane of Gram-negative bacteria, which belongs to the pathogen-associated molecular patterns (PAMPs). LPS stimulates host defensive reactions by binding to Toll-like receptor 4 (TLR4) [7], a member of the pattern recognition receptors (PRRs) typically expressed on leukocytes. According to their role in inflammation, both human and mouse platelets express TLR4, together with TLR2 and TLR9, implying that these cells may be responsive to LPS [10,11].

The ability of LPS to promote platelet activation in vitro has been widely debated in the past few years, and controversial results have been reported. Some data are consistent with LPS playing no role in platelet activation, but others suggest LPS as a genuine platelet agonist or a primer for platelet activation. Conversely, some studies have proposed LPS having a mild inhibitory effect on platelet activation induced by low doses of classical agonists [12,13,14,15,16,17,18,19,20].

In this review, we summarize the possible role of LPS in platelet activation with the aim of correlating the conflicting experimental evidence with the specific methodologies and reagents used, including the different procedures for platelet purification, doses and quality of LPS used, time of stimulation, and platelet functional readout. We critically examine the results obtained in the reported studies in terms of the ability of LPS to promote platelet aggregation, secretion, formation of reactive ox-ygen species (ROS) and interact with leukocytes. In 2017, Vallance and collaborators analyzed the contribution of LPS in platelet activation with a specific focus on the activated signaling pathways, and we refer to this paper for further information about this specific aspect [21].

## 2. The Importance of CD14 in Mediating LPS Binding to Platelet TLR4

LPS is a large molecule composed of lipid and sugar moieties, with a highly variable structure and molecular mass among different Gram-negative bacteria, depending on the composition and complexity of the oligosaccharide chains. The most commonly described LPS structure refers to that of *E. coli* and is composed of three distinct parts: the lipid A moiety, the core oligosaccharide, and the *O*-antigen polysaccharide (Figure 1). The lipid A moiety is localized into the outer membranes of the Gram-negative bacteria and is responsible for receptor recognition and binding [21,22]. It is composed of a highly acylated β-1′-6-linked glucosamine disaccharide that, in *E. coli*, is also phosphorylated. The distal glucosamine residue contains two secondary acyl chains, making the lipid A moiety of *E. coli* hexa-acylated. The core oligosaccharide is a linear or branched structure that links the lipid A moiety to the *O*-antigen, mostly composed of hexoses, heptoses, and octoses that can be modified by conjugation with phosphates or other substituents such as phosphoethanolamine [23]. The *O*-antigen represents a shield that prevents antibody recognition and is the most variable part of the LPS structure, generally assembled by the repeating of oligosaccharides composed of two to eight sugars in linear or branched structures [21,22,24]. In *E. coli*, N-acetylglucosamine-phosphate mannose, galactose, and rhamnose represent common sugars of the *O*-antigen oligosaccharide [25].

The functional association of LPS with its receptor TLR4 requires two accessory proteins: LPS-binding protein (LBP) and CD14. LBP is a 60 kDa glycoprotein synthesized by hepatocytes and released into the bloodstream at concentrations of 5–15 μg/mL [26]. LBP binding to LPS in vivo occurs preferentially within those microenvironments presenting a high density of LPS molecules, such as Gram-negative bacterial walls or LPS aggregates, and promotes the exposition of the lipid A moiety for subsequent interaction with the accessory protein CD14. When purified LPS is used for in vitro experiments, lipid A is naturally freely accessible, and thus, LPB binding is no longer required for LPS’ interaction with its receptor. Additionally, some observations in LBP-deficient mice have demonstrated that even in vivo LBP activity may be dispensable [21,27]. Finally, there is evidence that high mobility group box-1 (HMGB1) may form a complex with LPS and synergize in the TLR4-mediated activation of immune cells [28].

The monocyte differentiation antigen CD14 is a 55 kDa glycosylphosphatidylinositol-anchored glycoprotein expressed by immune cells. CD14 is associated with the cell membrane and, upon shedding, can release a soluble fragment of 13 kDa (sCD14), which is found in the plasma at concentrations of 1–3 µg/mL. CD14 is important for LPS’ presentation to TLR4 [21,27,29]. The N-terminal hydrophobic pocket of CD14 is able to bind up to five LPS molecules to generate an intermediate ternary complex [29]. Studies with CD14-deficient mice, however, demonstrated that, in the presence of high levels of LPS, as it occurs in an acute and rapid infection, the contribution of CD14 in promoting the binding of LPS to TLR4 is negligible [27].

TLR4 is composed of three domains. The extracellular domain is horseshoe-shaped and presents several leucine-rich repeats. The central domain is made of a single transmembrane hydrophobic α-helix, connected to the intracellular domain, which includes a Toll/IL-1 receptor (TIR) domain and mediates signal transduction. On the cell membrane, TLR4 is present as a heterodimeric complex in association with the myeloid differentiation factor 2 (MD-2), which is required for the binding of LPS. Thus, LPS-induced cell activation occurs upon the formation of a macromolecular LPS–CD14–MD-2–TLR4 complex [21] (Figure 2).

LPS binding to TLR4 stimulates two distinct signaling pathways: myeloid differentiation factor-88 (MyD88)-dependent and MyD88-independent pathways [30,31].

The MyD88-dependent pathway involves the interaction between the cytosolic domain of activated TLR4 and the TIR domain-containing adapter protein (TIRAP). Subsequently, the adaptor protein MyD88 binds the TIR domain of TIRAP and mediates the recruitment of interleukin-1 receptor-associated kinase 4 and 2 (IRAK4 and IRAK2), thus generating a macromolecular complex called the Myddosome, which promotes the phosphorylation of downstream targets including mitogen-activated protein kinases (MAPKs) Erk1/2 and p38MAPK, and nuclear factor of kappa light polypeptide gene enhancer in B-cells inhibitor, alpha (IκBα). This pathway eventually culminates with the activation of the transcription factor activator protein 1 (AP-1), and of nuclear factor kappa-light-chain-enhancer of activated B cells (NF-κB), which regulates the synthesis of type I interferon and other proinflammatory cytokines. The MyD88-independent pathway is initiated by the binding of the activated TLR4 with the adaptor proteins TIR-domain-containing adapter-inducing interferon-β (TRIF) and TRIF-related adaptor molecule (TRAM), which subsequently lead to the activation of the transcriptional factor interferon regulatory factor 3 (IRF3), responsible for the synthesis of proinflammatory cytokines [21,30] (Figure 2).

TLR4 is also expressed in platelets, which, therefore, are considered potential targets of LPS [12]. The modulation of TLR4 expression in platelets is controversial: some studies have reported that in vitro platelet stimulation with classical agonists (i.e., thrombin, convulxin, ADP, and epinephrine) increases TLR4 expression, possibly as a consequence of granule secretion [12]. However, this observation was not confirmed in other works [19,32]. For instance, Claushuis et al. demonstrated that the expression of platelet TLR4 was not enhanced in PRP stimulated with *Klebsiella pneumoniae* LPS, in mice infected with *K. pneumoniae*, and in patients with sepsis [32].

Shashkin et al. demonstrated that CD14 is necessary to induce LPS-mediated platelet responses in vitro. In this study, LPS triggered the activation of washed platelets exclusively in the presence of recombinant CD14 or small amounts of serum as a source of sCD14 [14]. Proteomic studies failed to detect the expression of CD14 in human platelets [33], but Berthet et al. demonstrated that platelets are able to take up sCD14 from exogenous sources, store it, and release it upon stimulation with LPS [20]. The observation that platelets do not express CD14 but can absorb a large amount of sCD14 from the plasma is also of physiological relevance, as platelets are not primary defenders in infection, but need to act in concert with immune cells that release sCD14 to trigger LPS–TLR4 signaling. Although platelet activation can involve the HMGB1 protein, and LPS has been shown to upregulate HMGB1 expression in platelets [34], it is currently uncertain whether, as for immune cells, HMGB1 plays any role in LPS-induced platelet activation.

## 3. Platelet Preparation Methods: Different Protocols Give Different Results

Platelet responsiveness can be assessed under three different environments: in whole blood (WB), in platelet-rich plasma (PRP), or in physiological buffers upon extensive washing procedures (WP). These experimental conditions are extremely different from each other and may significantly impact the outcome of the analysis, thus contributing to the generation of contradictory results [35,36]. The methodology used for platelet preparation is thus an important issue for the experimental design of LPS-induced platelet activation [12].

Platelets in WB undergo minimal manipulation, and events that may affect platelet responsiveness are related to the modality and the time of blood withdrawal, the type of needle, and the anticoagulant used [36]. Nevertheless, because of the presence of plasma and other blood cells, the analysis of platelets in WB has important experimental limitations. As a matter of fact, WB is almost exclusively used for the analysis of platelet function by flow cytometry. The use of fluorochrome-conjugated antibodies directed against platelet-specific antigens (e.g., CD41: integrin αIIb, and CD42b: glycoprotein Ibα) allows one to discriminate the platelet population from white blood cells (WBCs) and red blood cells (RBCs). Platelet activation is commonly measured in the gated population by analyzing the exposure on the platelet surface of CD62P and CD63, markers of α-granule and δ-granule release, respectively, or by analyzing integrin αIIbβ3 activation using the epitope-specific antibodies PAC-1 (human) and JON-A (mouse) [37,38,39]. Additionally, other parameters measurable with specific fluorometric probes (such as ROS production, mitochondrial vitality, intracellular Ca^2+^, and VASP phosphorylation) can be investigated in platelets in WB.

PRP is obtained from anticoagulated blood upon a single centrifugation step, typically at 250× *g* for 13–16 min, and allows one to discharge both RBCs and WBCs while preserving the platelets in plasma. PRP contains plasma proteins (including coagulation factors) and ions at physiological concentrations and is suitable for studying platelet behavior without interference from other blood cells. However, the time, temperature, and speed of the centrifugation may affect platelet activation, resulting in desensitization [35,36].

WP is obtained from PRP by subsequent centrifugation and washing steps in physiological buffers. Typically, three washing steps are recommended to eliminate any trace of plasma, although several reported protocols are less rigorous, and, in some cases, a single washing step is reported. It has been demonstrated that centrifugation may preactivate platelets, as resting WP, but not PRP, expresses some levels of CD62P and activated integrin αIIbβ3 on the platelet surface [12,40]. Moreover, RBCs release ADP during centrifugation steps, which induces platelet desensitization [18]. To minimize these effects, platelet inhibitors are typically added during the preparation procedure, including ADP scavengers and cyclooxygenase inhibitors [35,41,42,43]. The contamination of WBCs in PRP and in WP can be evaluated through flow cytometry using anti-CD3, anti-CD14, anti-CD15, anti-CD45 and anti-CD19 antibodies [20,44].

It is clear that WP is heavily manipulated during the preparation procedure, and this can alter the natural responsiveness. Nevertheless, the use of WB has some important advantages, as it allows experimental approaches not feasible with other kinds of preparations, without interference by the anticoagulant added or by plasma components. For instance, some traces of thrombin can be generated during blood collection and PRP preparation. Furthermore, the washing procedure allows the recovery of platelets in an artificial medium with controlled concentrations of ions, proteins, and other molecules [35,41].

In the analysis of LPS-induced platelet activation, the use of WB, PRP, or WP may be particularly critical, because of the presence of other blood cells or plasma proteins (e.g., CD14). Most of the studies on the effect of LPS on platelet activation currently available have been performed using WP. The use of PRP is less common, and analyses on WB are very scarce. The predominant use of WP justifies the fact that LPS is often added in combination with recombinant CD14 and/or LBP [14,20,44,45], although this approach is not pursued in all the available studies. The addition of recombinant CD14 along with LPS has been reported to elicit responses in WP comparable to those observed in PRP [20,44]. The different methods of platelet preparation and a brief description of the platelet response to LPS are summarized in Table 1 and detailed throughout the text.

## 4. Sources of LPS, Doses, and Time of Stimulation

A critical factor influencing the outcome of the analysis of LPS-induced platelet activation is certainly the source of the reagent. The most common form of LPS used in the literature is LPS O111:B4 from *E. coli* [12,13,15,16,17,18,20,32,44,46,48], but other bacterial species or serotypes have been considered, such as *E. coli* LPS O127:B8, O55:B5 [15], and K12 [13], *Salmonella minnesota* LPS [20,46], *K. pneumoniae* LPS, and *Pseudomonas aeruginosa* LPS [32]. Moreover, in some studies, the bacterial species or the strain [14,15,19,32,45,50] was not specified. LPS from different species or serotypes displays different potency in eliciting some responses in platelets [13,15,20], including ROS production [13], platelet aggregation, cGMP production [15], δ-granule secretion, and PDGF-AB release [20]. Besides LPS of different bacterial species, Kappelmayer et al. have demonstrated that different forms of LPS may have distinct effects on platelet activation and coagulation. A heterogeneous mixture of LPS with varying numbers of repeating polysaccharides is present on bacteria, together with a smooth form of LPS (S-LPS), which comprises a lipid A moiety, core-oligosaccharide, and *O*-polysaccharides, and the rough form of LPS (Re-LPS), lacking *O*-specific chains. Interestingly, the stimulation of platelets with Re-LPS, but not with S-LPS, directly modulates platelet activation and coagulation, and increases TRAP-induced aggregation and CD40L exposure [51].

Additionally, potential contaminants of LPS preparations, in particular, additional components of the bacterial cell wall, may stimulate TLR2 [12,52]. The effect of ultrapure LPS from diverse bacterial species on the modulation of platelet activation has been extensively investigated [46]. Ultrapure LPS does not increase α-granule secretion and ROS production per se and does not increase ADP/collagen/TRAP6-induced aggregation [46].

The pathological range of concentration of LPS detectable in the blood of patients with sepsis associated with community-acquired pneumonia is 88–131 pg/mL (107 pg/mL on average), compared to 9–15 pg/mL (13 pg/mL on average) for healthy controls [17]. The concentrations of LPS used for in vitro experiments are extremely variable, ranging from 15 pg/mL [17] to 100 μg/mL [12,15,18], although, in the majority of the studies, LPS is used in supraphysiological concentrations in a range between 1 and 10 μg/mL [10,12,13,15,16,18,19,20,32,44,46,47,48,49,50]. This choice could reflect the fact that, in vivo, LPS stimulates, at the same time, immune cells and platelets, thus potentiating the final effects, whereas in vitro or ex vivo, on isolated cells, the minimal concentration required to obtain the same effect is higher. However, it is important to note that at least one recent study used physiological concentrations of LPS, as detected in the plasma of patients with sepsis, and demonstrated that LPS potentiates agonist-induced platelet aggregation [17].

The time of platelet stimulation with LPS may elicit different platelet responses. Most of the studies report times of stimulation ranging from 10 to 30 min [10,12,13,15,16,17,18,19,20,32,44,46,47,48,50]. This time of incubation is unusually long for the evocation of classical platelet responses, but it appears physiologically reasonable, as LPS does not act as a classical agonist that directly activates hemostatic and thrombotic responses; it rather represents a warning signal for infections.

In the light of the considerations on the factors that can influence data collection reported above, in the next paragraphs, we critically summarize the results obtained in the currently available studies that have used LPS to activate platelets in vitro. The results are also summarized in Table 1.

## 5. Effect of LPS on Platelet Activation

### 5.1. LPS Promotes Maturation and Release of IL-1β

Although platelets are anucleated cellular fragments, they retain mRNA, ribosomes, and the ability to synthetize bioactive enzymes and mediators of inflammation [53] and immunity [54]. In particular, activated platelets synthetize COX, which generates inflammatory mediators, and IL-1β [53], and release proinflammatory cytokines and chemokines.

IL-1β and COX2 pre-mRNA splicing and synthesis in platelets were also described upon stimulation with LPS. The induction of IL-1β synthesis by LPS was analyzed by multiple approaches, including real-time PCR, flow cytometry, and confocal microscopy, on a population of highly purified isolated platelets. To exclude the possible contamination of monocytes, isolated platelets underwent a two-step purification protocol obtained by immunodepletion with a combination of anti-CD14, anti-CD15, and anti-CD45 conjugated to magnetic beads. Anti-glycophorin was used to remove residual erythrocytes. The contamination of monocytes was confirmed to be negligible (a few per million platelets). In this experimental setting, LPS, in the presence of autologous plasma or recombinant CD14 and LBP, induced IL-1β synthesis upon 3 to 4 h of stimulation [14]. LPS promoted the maturation and secretion of IL-1β in specific microvesicles through the binding to TLR4 and the initiation of a signaling cascade involving phosphoinositide 3-kinase (PI3K)/Akt and c-Jun N-terminal kinase (JNK) [45] (Figure 3). LPS was found to be more potent than thrombin in inducing the maturation of IL-1β, although, in the same studies, it did not initiate classical platelet responses. These observations suggest that LPS contributes to inflammatory responses by activating signaling pathways alternative to those required for hemostasis. IL-1β released by LPS stimulation is able to activate platelets through an autocrine stimulatory loop [55], and to activate endothelial cells [45].

LPS also regulates the release of additional inflammatory mediators from platelets. For instance, some studies have reported that LPS promotes the release of sCD40L, platelet factor 4 (PF4), platelet-activating factor (PAF), RANTES, and angiogenin when added to PRP or also to WP if stimulation is performed in the presence of recombinant CD14 [19,20,44]. However, a recent study demonstrated that LPS promotes the release of RANTES, PDGF, and PF4 in PRP and WP even in the absence of CD14. Moreover, the release of sCD40L and neutrophil-activated peptide-2 was detected exclusively in WP, but not in PRP [12].

### 5.2. LPS in Platelet–Leukocyte Aggregate Formation and NETosis

Platelet activation initiates a crosstalk with WBCs, inducing the formation of platelet–leukocyte aggregates (PLA) through specific ligand/receptor binding. In particular, platelet CD62P and CD40L bind to leukocyte P-selectin glycoprotein ligand-1 (PSGL1) and CD40. PLA are considered an important index of the inflammatory state, as their number in the circulation increases during inflammation, infections, and sepsis [56]. Platelet–neutrophil aggregates (PNA) and platelet–monocyte aggregates (PMA) are the most abundant types of circulating PLA [57,58]. As a result of interaction with platelets, neutrophils become activated and release neutrophil extracellular traps (NETs): DNA-based structures acting as traps to capture and neutralize the pathogens [59]. However, an exaggerated NET production is involved in endothelium injury and inflammation [60].

In a model of sepsis in vivo, in which mice are intraperitoneally injected with LPS, NETosis is enhanced [61,62,63,64], and NETs are also elevated in septic patients [65,66]. In an in vitro model, platelets activated by LPS, but not resting platelets, bind to adherent neutrophils under flow conditions. This effect is mediated by platelet TLR4, as it is completely inhibited by a TLR4-specific antibody [48]. In a similar experimental setting, LPS-activated platelets are also able to induce NETosis [48]. The methodology for neutrophil preparation is unlikely to contribute to cell activation, as NETosis was not observed in the absence of stimulated platelets. Although there is no doubt that LPS-activated but not resting platelets stimulate NETosis, it is not clear whether the TLR4–LPS complex itself formed on the platelet surface is directly responsible for neutrophil TLR4 stimulation able to trigger NETosis or whether this is a result of other events triggered by LPS stimulation in platelets. However, there is direct evidence that the LPS-induced crosstalk between platelets and neutrophils is driven by the ability of LPS to promote platelet CD62P exposure and consequent interaction with PSGL1 on neutrophils [14].

More recent studies using WP and quiescent purified neutrophils, however, failed to demonstrate that LPS can stimulate PNA formation or increase PNA formation induced by the classical platelet agonist thrombin [16]. Moreover, PNA or PMA formation was not detected even when LPS was added to WB [32,47].

### 5.3. Role of LPS in Platelet Granule Secretion

α-granules and δ-granules are specific platelet organelles containing adhesion and repairing factors (e.g., PF4, fibronectin, fibrinogen, IL-1β, and vWF) and pro-aggregating factors (e.g., ADP, ATP, serotonin, and calcium), respectively. They also differ in membrane components. α-granules express CD62P, whereas δ-granules expose CD63. Granule secretion occurs upon platelet activation induced by most of the physiological agonists [67].

α- and δ-granule secretion induced by LPS have been extensively analyzed by flow cytometry (CD62P, CD40L, and CD63 expression) [10,13,14,15,16,18,19,20,32,44,46,47,48,49,50], ELISA (vWF release), [16] and measuring ATP release [15,16] or mepacrine fluorescence [18]. It has been reproducibly observed that LPS potentiates granule secretion induced by thrombin [16], U46619 [13], or collagen-related peptide (CRP) [46]. However, the ability of LPS alone to stimulate granule secretion is highly controversial. Some studies have reported that LPS promotes α-granule and/or δ-granule secretion in PRP [13,19,20,44,50], as well as in WP, both in the presence of recombinant CD14 [14,20,44] and without added CD14 [15,16]. However, other studies failed to detect any secretion of both α- [10,18,20,32,46,47,49] and δ-granules [18,32,47] in LPS-stimulated platelets. The reason for such an inconsistency in the published observations is unknown.

### 5.4. LPS Promotes ROS Generation

The generation of ROS by activated platelets is critical during inflammation, as ROS are able to activate platelets and regulate the function of surface receptors required for platelet–neutrophil interactions during vascular inflammation [68,69].

The generation of ROS by platelets was documented exclusively in response to high concentrations of LPS [17] (1–10 μg/mL) both in WP [13] and PRP [50], but was not observed upon stimulation with LPS in a lower range of concentrations (15–100 pg/mL). Additionally, LPS increased ROS production in WP stimulated with low doses of the thromboxane analog U46619, CRP [13], collagen, and ADP [17] (Figure 3). When added to PRP, LPS induces platelet mitochondrial depolarization, an early effect of apoptosis related to ROS production and, thus, platelet damage [50].

### 5.5. LPS Is Involved in the Activation of Canonical and Non-Canonical Signaling Pathways

Typically, the stimulation of TLR4 leads to the activation of transcriptional factors and thus gene regulation. The downstream effectors and signaling pathways stimulated by TLR4 in platelets are currently under investigation [13,16,45,50]. Since platelets lack nuclei, transcription factors activated downstream of TLR4, such as NF-κB, are supposed to exert atypical non-genomic events. The signaling pathways triggered by LPS binding to TLR4 are reviewed by Wallance et al. [21].

The activation of signaling cascades downstream of TLR4 is barely detectable in WP when stimulation with LPS is performed in the absence of added CD14 [13,17]. However, in the presence of CD14 and LBP, LPS is able to stimulate the phosphorylation of p38 MAPK, JNK, and Akt, a downstream effector of PI3K. The phosphorylation of intracellular kinases and effectors has been analyzed by immunoblotting using phospsho-specific antibodies [13,16,17,50] and by ELISA [45]. As mentioned above, LPS activates platelet TNF-receptor-associated factor 6, MyD88, TIRAP, and IRAK1/4, and these effectors are required for LPS-induced IL-1β mRNA maturation, splicing, and IL-1β release [45]. Interestingly, LPS binding to TLR4 potentiates platelet activation induced by GPVI agonists or by U46619 through the activation of the PI3K pathway, followed by increased phosphorylation of Akt and ERK1/2 [13]. Moreover, the interaction of LPS with TLR4 increases collagen and ADP-induced NADPH oxidase activation, ROS production, and TxA2 synthesis through the activation of cPLA_2_ [17] (Figure 3).

LPS promotes NF-κB activation through the degradation of the NF-κB phosphorylation inhibitor, IκBα, and the phosphorylation of p65, a subunit of NF-κB [16] (Figure 3). Stimulation with LPS in the presence of thrombin significantly increases IκBα degradation and p65 phosphorylation. The activation of NF-κB has been proposed as essential to mediate the platelet aggregation and granule secretion induced by LPS [16].

### 5.6. LPS’ Effects on Integrin αIIbβ3 Activation

Activated integrin αIIbβ3 binds fibrinogen, vWF, fibronectin, and vitronectin and is critically involved in platelet aggregation [70].

*E. coli* LPS O111:B4, but not K12, promotes integrin αIIbβ3 activation in PRP [13]. This effect is dependent on the presence of plasma and is not observed in WP [16,18]. However, in WP, LPS increases the integrin αIIbβ3 activation stimulated by classical agonists such as thrombin [16], U46619, and CRP [13]. The pretreatment of platelets with LPS also increases the adhesion of platelets to fibrinogen under low-shear-flow conditions, and this effect is dependent on LPS binding to TLR4 [10].

### 5.7. LPS and Platelet Aggregation

Septic patients manifest increased platelet aggregation [6,71] and develop disseminated intravascular coagulation in several organs due to the formation of microthrombi [66]. In vitro platelet aggregation is barely reproducible by the sole addition of purified LPS. In all the published studies, there is a general consensus regarding the inability of LPS to promote platelet aggregation per se both in PRP and in WP [12,13,14,15,16,17,48]. Only a single study reported the direct ability of LPS to promote platelet aggregation in PRP via interaction with TLR4 [50]. More interestingly, the majority of available publications have demonstrated that LPS potentiates platelet aggregation induced by low doses of thrombin [12,15,16], collagen/CRP [13,15,17], ADP [14,17], and U46619 [13]. Based on the observation that the stimulation of platelets with classical agonists increases the expression of TLR4 on the platelet surface [20,72], Vallance and collaborators tested the possibility that the preincubation of platelets with ADP, collagen, or TRAP6 might increase the aggregation induced by ultrapure LPS. In this experimental setting, however, they failed to observe any significant variation of aggregation [46].

Martyanov and collaborators recently analyzed platelet aggregation induced by LPS in hirudinated PRP. They demonstrated that LPS inhibits the reversible platelet aggregation elicited by low doses of ADP and TRAP6, but not CRP. This action was proposed to be mediated by the inhibition of phosphodiesterase and, thus, the increase in cytosolic cAMP/cGMP concentration and protein kinase A/G activation. Conversely, LPS was capable of compensating for ADP-induced P2-receptor desensitization in WB, restoring the calcium mobilization and fibrinogen binding. This observation might explain the inconsistency in previous data, as ADP is released by RBCs during the platelet-purification process [18].

## 6. Discussion

The role of LPS in platelet activation in vitro has been widely analyzed in the last two decades, but several questions remain unsolved, mainly because of the diversity of the experimental approaches used. First, the use of WB, PRP, or WP introduces considerable differences in platelet preactivation/desensitization, due to the different manipulation [12,18,35,36,41,42,73]. Second, the presence of CD14 and/or LBP or plasma/serum may be critical for eliciting the direct effects of LPS in platelets [14,20,44,45]. Additionally, plasma proteins should be considered as important modulators of chemokine levels, interfering with the cleavage of endopeptidase or protein binding [12], as indicated by independent reports [74,75]. Third, the LPS purity, LPS concentration, and time of stimulation of platelets are important for highlighting a specific effect of LPS [14,17,20,46]. Fourth, different LPS sources and strains promote different platelet activation [13,15,20,32,46,51].

The investigation of platelet activation induced by LPS in vitro does not recapitulate the complexity that occurs in vivo during sepsis. In pathological conditions, different cells are simultaneously involved and are activated by multiple stimuli. Platelets, for instance, are activated directly and indirectly by interaction with toxins and through crosstalk with other blood cells, such as monocytes and neutrophils [6,21,71,76,77]. The concentration of circulating LPS during sepsis cannot be reliably correlated with the concentration of Gram-negative bacteria, mainly because the size and composition of LPS are highly dynamic among bacterial species. Moreover, there is no doubt that immune cells are more reactive than platelets to LPS even at low concentration. Nevertheless, the impact of platelets in sepsis is certainly relevant and cannot be neglected. In mice injected with LPS, and in patients with sepsis, the platelet count drops rapidly, and the resulting thrombocytopenia is correlated with the worst prognosis and poor outcomes [78].

However, despite several limitations and considering the cautions that we have reported in this review, the analysis of LPS-induced platelet activation in vitro provides some important clues. In particular, it helps one to understand the specific molecular mechanisms and signaling pathways triggered by LPS binding to TLR4, allowing one to decipher the hierarchy of LPS signaling in platelets and to identify crucial intracellular effectors, which could be of therapeutical interest in the treatment of thromboinflammation and sepsis.

In this context, however, some questions remain open. First, it is not clear whether LPS interacts solely with TLR4. Some researchers, in fact, have proposed a combined activity of TLR2 and TLR4 in LPS recognition and signaling [79,80], whereas others have observed that the inflammatory caspases are innate immune receptors for cytosolic LPS [81,82]. The possibility that LPS triggers platelet activation through the combined action of different types of receptors is intriguing and has not been adequately investigated to date. Second, the intracellular signaling pathways stimulated by LPS in platelets are still poorly characterized [21]. Little information is available on the MyD88-independent pathway in platelets, despite its importance in LPS-induced signaling. In particular, the expression of the transcription factor IRF3 in platelets and its possible role in their activation have never been clarified.

In conclusion, LPS has a significant but not entirely clarified role in platelet activation and platelet-derived inflammation. The use of specific pharmacological inhibitors in vitro and the generation/use of transgenic mice will allow researchers to better define the role of signaling proteins in LPS-induced platelet responses, increasing the understanding of this process and possibly leading to the identification of novel players. These future studies will be essential to allow the identification of potential targets for the development of new strategies for the modulation of thromboinflammation and sepsis.

## Figures and Tables

**Figure 1 ijms-23-10900-f001:**
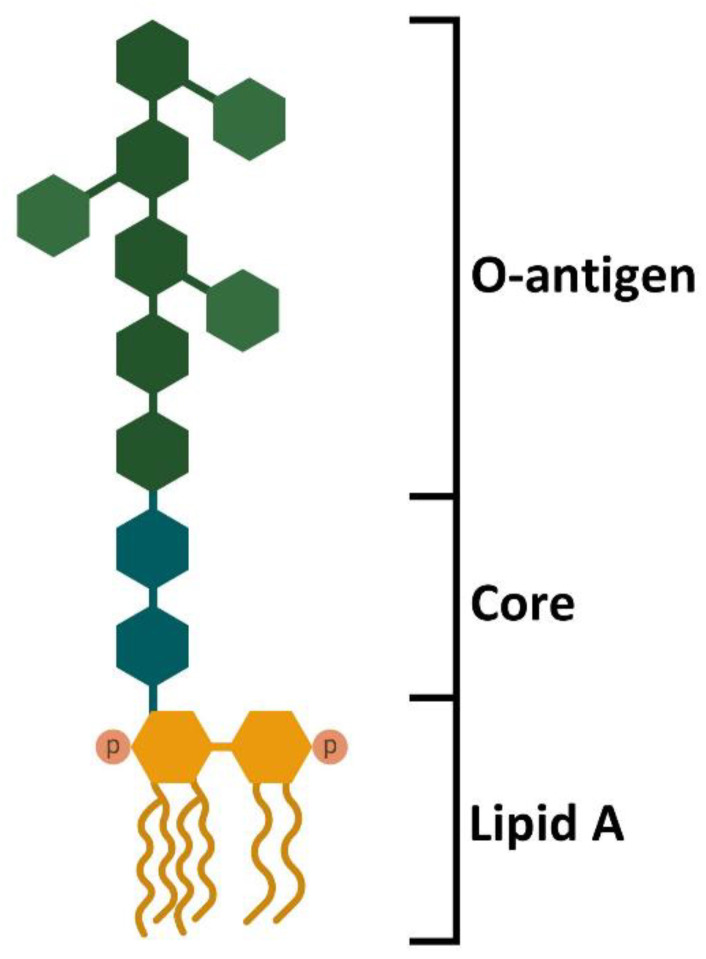
Structure of *E. coli* LPS. LPS is a complex molecule made of lipids and sugars. The structure is divided into three different domains. The lipid A moiety is localized into the outer membrane of the Gram-negative bacteria and is responsible for TLR4 interaction through the phosphate groups and the acyl chains. The central core is composed of oligosaccharide and links the lipid A moiety with the *O*-antigen. The *O*-antigen is made of polysaccharides and can reach over 100 repeats of sugars. The complexity and the high number of repeats allow a more stringent barrier against antibody recognition. Furthermore, the *O*-antigen allows differentiating bacterial species and serotypes [21,22].

**Figure 2 ijms-23-10900-f002:**
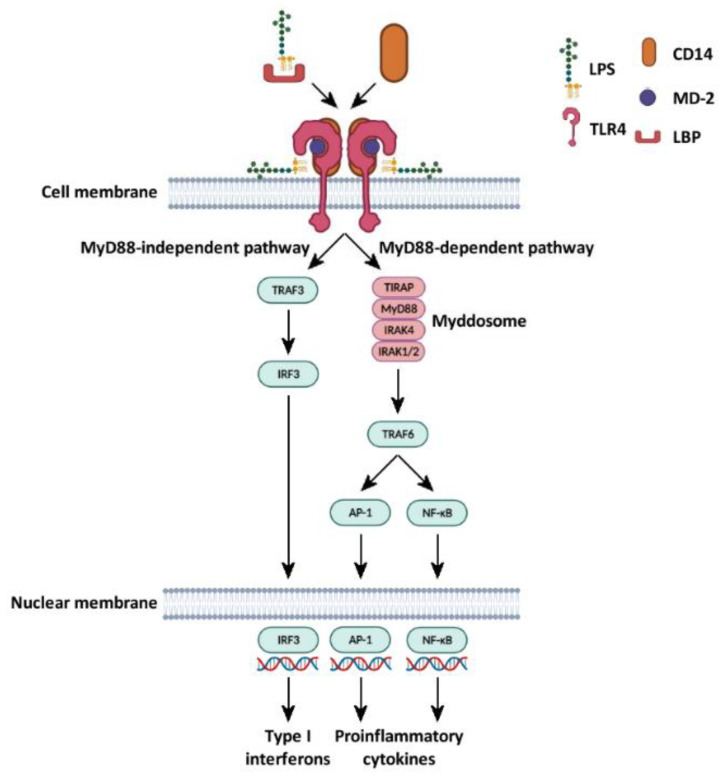
LPS–TLR4 signaling cascades in nucleated cells. LPS-binding protein (LBP) binds to LPS, inducing the exposure of the lipid A moiety and acting as a lipid-transfer protein. LBP drives the interaction of LPS with CD14, a glycosylphosphatidyl inositol-anchored glycoprotein. CD14 acts as a monocyte membrane coreceptor, but it can be released as a soluble protein. The interaction with CD14 allows the transfer of LPS to myeloid differentiation factor 2 (MD-2), a protein that interacts and cooperates with TLR4, activating the signal transduction. MyD88-dependent pathway involves MyD88, TIRAP, IRAK4, and IRAK 1/2, constituting the Myddosome. It activates TRAF6, which in turn activates two transcription factors (AP-1 and NF-κB) that allow the expression of proinflammatory cytokines. MyD88-independent pathway does not involve MyD88, but activates TRAF3, which in turn activates IRF3, another transcription factor, involved in type I interferon expression [21,22,27,29].

**Figure 3 ijms-23-10900-f003:**
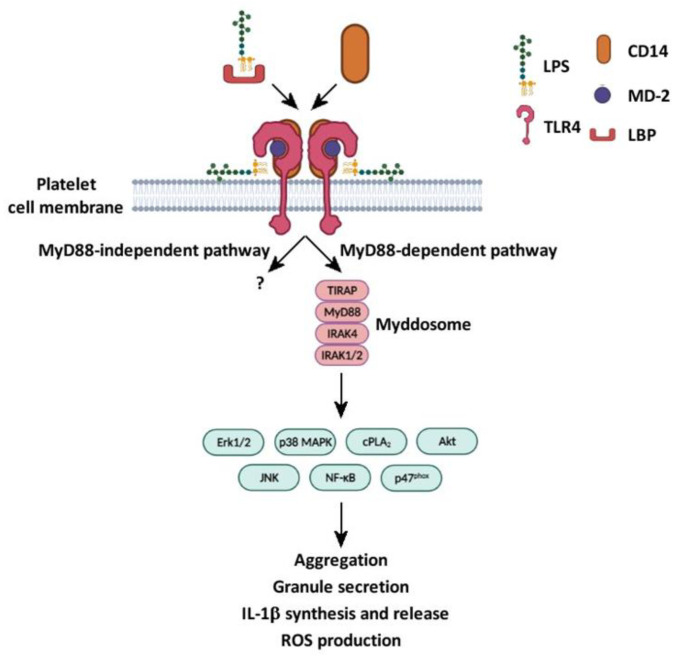
LPS–TLR4 signaling cascades in platelets. LPS initiates the MyD88-dependent signaling cascade, which in turn activates many downstream effectors (Erk1/2, p38 MAPK, cPLA_2_, Akt, JNK, NF-κB, and p47^phox^), leading to or potentiating platelet aggregation, granule secretion, IL-1β synthesis and release, and ROS production [10,12,13,14,15,16,17,19,20,21,44,45,50]. The role of MyD88-independent pathway is still unclear.

**Table 1 ijms-23-10900-t001:** Materials, methods, and results in LPS-induced platelet activation experiments. In the table are summarized the main results and the main different experimental conditions among the studies: LPS sources/strains, platelet preparation, LPS and CD14/LBP concentrations, and time of stimulation of platelets.

LPS Source/Strain	Platelet Preparation	LPS (µg/mL)	CD14/LBP (µg/mL)	Time (min)	Results	Ref.
*E. coli*/ O111:B4	WP	5	Serum	30	LPS promotes platelet binding to fibrinogen but not α-granule secretion	[10]
*E. coli/* O111:B4 *S. minnesota/* R595	WB PRP WP	0.125–2	No	20–25	LPS does not increase α-granule secretion and ROS production both per se and by CRP. LPS does not increase ADP/collagen/TRAP6-induced aggregation	[46]
*E. coli* *K. pneumoniae* *P. aeruginosa*	WB PRP WP	0.1–5	No	10–90	LPS does not increase α- and δ-granule secretion, PS exposition, and PNA and PMA formation per se and by ADP/CRP	[32]
*E. coli*	WB	5	No	30	LPS does not induce α- and δ-granule secretion and PLA formation	[47]
*E. coli*/ O111:B4	WP	5	No	10	LPS induces PNA but not NET formation, α-granule secretion, and aggregation per se	[48]
*E. coli*/ O111:B4	PRP	1	No	60	LPS does not promote α-granule secretion and aggregation per se	[49]
*E. coli*/ O111:B4	PRP WP	10	No	30–120	LPS does not promote granule secretion, fibrinogen binding, and PS exposure. LPS decreases ADP-induced aggregation in hirudinated PRP	[18]
*E. coli*/ O111:B4; K12	PRP WP	0.5–7.5	No	5–10	LPS promotes α-granule secretion and integrin activation and potentiates U46619/CRP activity. LPS does not induce ROS production and aggregation per se, but both are potentiated by U46619 and CRP	[13]
*E. coli*/ O111:B4	WP	0.5–10	No	15	LPS induces vWF release. LPS does not promote ADP release and aggregation per se, but both are potentiated by thrombin	[16]
*E. coli*/ O111:B4	PRP WP	15	No	30	LPS induces RANTES, PDGF, and PF4 release in PRP and WP, but NAP-2 and sCD40L release only in WP. LPS potentiates thrombin-induced aggregation in WP	[12]
	PRP WP	10	No	30	LPS promotes α-granule secretion, ROS production, and aggregation per se	[50]
*E. coli*/ O111:B4	PRP WP	1–10	0.25/No	30	LPS promotes α- and δ-granule secretion	[44]
*E. coli/* O111:B4 *S. minnesota*	PRP WP	3	1/No	30	LPS induces δ- but not α-granule secretion. LPS promotes sCD40L and sCD14, and inhibits RANTES and PDGF-AB	[20]
*E. coli*	WP	0.1	0.1/ 0.1	180	LPS promotes IL-1β maturation and release	[45]
*E. coli/* O111:B4; O127:B8; O55:B5	WP	1	No	10–30	LPS promotes α-granule and ADP secretion. LPS does not induce aggregation per se but potentiates thrombin/collagen-induced aggregation	[15]
*E. coli*	WP	0.1	0.15/ 0.1 Serum	60–180	LPS promotes IL-1β maturation and release, PNA formation, and α-granule secretion. LPS does not induce aggregation per se but potentiates ADP-induced aggregation	[14]
*E. coli*	PRP	0.5–2	No	15	LPS promotes δ-granule secretion. LPS induces sCD40L and PAF4, and inhibits RANTES, PDGF-AB, and angiogenin	[19]
*E. coli*/ O111:B4	WP	<0.1	No	15	LPS does not induce aggregation per se but potentiates ADP/collagen-induced aggregation TxA2 and ROS production	[17]

## Data Availability

Not applicable.

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
