# Peer review of "The Controversial Role of LPS in Platelet Activation In Vitro"

_ijms, 2022, doi:10.3390/ijms231810900_

Round 1
Reviewer 1 Report
In this review, the authors analyzed the effect of LPS on platelets. In particular, they reviewed several studies about LPS-induced platelet activation in vitro discussing different methodological aspects and LPS-mediated effects in terms of aggregation, granule secretion, cytokine release, ROS production, and interaction with leukocytes and NET formation.
The review is very interesting, well-written, and well organized. The results of the analyzed studies are well described in Table 1 and the figures are appropriate.
The effect of LPS on platelets depends on several factors which influence the platelet response and are well described in the paper. In this regard, of particular importance, for example, are the questions concerning the role of CD14, methods of platelet preparations and LPS concentration.
I have only two considerations:
1) As can be seen from Table 1, the LPS concentrations used in vitro are supraphysiological. The authors could better discuss this point and how it affects platelet activation.
2) LPS activates TLR4. Authors should report studies supporting TLR4 activation by LPS.
Author Response
In this review, the authors analyzed the effect of LPS on platelets. In particular, they reviewed several studies about LPS-induced platelet activation in vitro discussing different methodological aspects and LPS-mediated effects in terms of aggregation, granule secretion, cytokine release, ROS production, and interaction with leukocytes and NET formation.
The review is very interesting, well-written, and well organized. The results of the analyzed studies are well described in Table 1 and the figures are appropriate.
The effect of LPS on platelets depends on several factors which influence the platelet response and are well described in the paper. In this regard, of particular importance, for example, are the questions concerning the role of CD14, methods of platelet preparations and LPS concentration.
I have only two considerations:
- As can be seen from Table 1, the LPS concentrations used in vitro are supraphysiological. The authors could better discuss this point and how it affects platelet activation.
We thank the reviewer for highlighting this point. The pathological range of LPS detectable in the blood of patients with sepsis is about 100 pg/ml whereas concentration in healthy subjects is around 10 pg/ml, as mentioned in section 4. The majority of the in vitro studies used LPS concentration between 1 to 10 micrograms/ml, which is 10 to 100 time higher than that potentially achievable in human circulation during sepsis. The use of supraphysiological concentrations is justified because in vitro experiments, especially studies on isolated platelets, cannot recapitulate the situation in vivo, in term of complexity and cell interactions. However high concentrations may also result in artefacts in platelet responses. Nocella et al. [17] studied the effect of LPS, in a range of concentrations detected in patients with sepsis secondary to community-acquired pneumonia and demonstrated that LPS potentiates agonist-induced platelet aggregation. This is now better specified in the text in section 4.
- LPS activates TLR4. Authors should report studies supporting TLR4 activation by LPS.
References that report studies on LPS binding to TLR4 has been added in session 2:
[30] Lu YC, Yeh WC, Ohashi PS. LPS/TLR4 signal transduction pathway. Cytokine. 2008;42:145-151.
[31] Pålsson-McDermott EM, O'Neill LA. Signal transduction by the lipopolysaccharide receptor, Toll-like receptor-4. Immunology. 2004;113:153-62.
Specific reference [46.Vallance et al., 2017] on TLR4 signaling in platelets is now more clearly indicated in section 5.5
Reviewer 2 Report
The review of Galgano et al. is devoted to the deciphering of the controversial effect of the lipopolysaccharides action on platelets. The authors provide accurate description of LPS structure, LPS sources and give an excellent overview of the present opinions in the field. However, correction/addition of several points could improve the quality of review.
Major comments
1. It has been proposed that TLR4-mediated platelet activation by LPS can be enhanced by the presence of HMGB1 protein, which forms complexes with LPS. Please discuss this in the corresponding section of the manuscript.
2. Besides LPS of different bacterial strains, Kappelmayer et al. have also reported that different forms of LPS also have distinct effects. Furthermore, different effects of LPS can also be justified by different LPS preparations: LPS molecules tend to form aggregates and this can result in the modulation of LPS activity.
3. It would be interesting to see the correspondence of the purified LPS concentration to the concentration of gram-negative bacteria in the case of sepsis. Furthermore, LPS effect on immune systems should be significantly more profound than on platelets in vivo. Hereby, please discuss whether you believe if LPS impact on platelets would be significant of negligible in vivo.
4. In the section 5.1 authors discuss LPS capability to induce IL1b synthesis by platelets. However, IL1b synthesis has been detected upon platelet storage for more than 24 hours, what should have had a significant impact on platelet state. Thus, please give additional details of the experimental design to the main text.
5. In the section 5.2 the authors discuss that LPS-activated platelets might induce NETosis. Please, discuss whether it is possible that LPS binds to platelet TLR4 and upon platelet interaction with neutrophils platelet TLR4-LPS complex can cause neutrophil TLR4 activation and subsequent NETosis? Finally, please give additional details on the procedures of neutrophil washing as it can significantly affect neutrophil activation.
6. In the section 5.5 please add information on the methods of detection of activation of different pathways of LPS-induced platelet activation.
7. In the discussion the authors claim that the use of pharmacologic inhibitors and/or transgenic mice can be of help for determination of the LPS role for platelet activation. It would be great to see more specifics on what authors propose.
Minor comment
1. Please avoid recurrent description of the LPS structure, LBP and sCD14 present in the section 2.
Author Response
The review of Galgano et al. is devoted to the deciphering of the controversial effect of the lipopolysaccharides action on platelets. The authors provide accurate description of LPS structure, LPS sources and give an excellent overview of the present opinions in the field. However, correction/addition of several points could improve the quality of review.
Major comments
1- It has been proposed that TLR4-mediated platelet activation by LPS can be enhanced by the presence of HMGB1 protein, which forms complexes with LPS. Please discuss this in the corresponding section of the manuscript.
It has been shown that High mobility group box-1 (HMGB1) works in synergy with LPS in activating immune cells [28.Yang et al., Targeting Inflammation Driven by HMGB1. Front Immunol. 2020], and enhances LPS activity in macrophages through the binding of TLR4 and the activation of MyD88-dependent pathway (Vogel et al, HMGB1 is a critical mediator of thrombosis. J Clin Invest. 2015).
A recent study demonstrated that surface expression of HMGB1 is enhanced in LPS-stimulated platelets (Li et al., Lipopolysaccharide-Activated Canine Platelets Upregulate High Mobility Group Box-1 via Toll-Like Receptor 4. Front Vet Sci. 2021), but, to our knowledge, whether this protein can influence LPS-induced platelet activation remains unclear. These notions have been now mentioned in section 2.
2-Besides LPS of different bacterial strains, Kappelmayer et al. have also reported that different forms of LPS also have distinct effects. Furthermore, different effects of LPS can also be justified by different LPS preparations: LPS molecules tend to form aggregates and this can result in the modulation of LPS activity.
We thank the reviewer for this suggestion. The text has been now updated in section 4 with information on the effects of different forms of LPS (smooth versus rough) on platelet activation.
3- It would be interesting to see the correspondence of the purified LPS concentration to the concentration of gram-negative bacteria in the case of sepsis. Furthermore, LPS effect on immune systems should be significantly more profound than on platelets in vivo. Hereby, please discuss whether you believe if LPS impact on platelets would be significant of negligible in vivo.
According to Yagupsky and Nolte (Yagupsky and Nolte. Quantitative aspects of septicemia. Clin Microbiol Rev. 1990) Gram-negative bacteremia consists in about 10 bacteria/ml of blood in adult individuals, which correspond to about 100 pg/ml (the pathological range of LPS detectable in the blood of patients with sepsis [17. Nocella et al., 2017], as reported in section 4). It is however difficult to correlate the number of bacteria with the expression of different form of LPS, as the size and composition of LPS are highly dynamic among bacterial species.
Regarding the effects of LPS on immune cells, the reviewer is absolutely right. Immune cells are the first player in immunity, and are primary activated by LPS, toxins or bacteria, whereas platelets can be activated either by pathogens or by pathogen-released compounds or by activated immune cells. However, the immunomodulatory role of platelets in the context of sepsis is becoming increasing evident (Assinger et al., Platelets in Sepsis: An Update on Experimental Models and Clinical Data. Front Immunol. 2019), as platelets are able to display an inflammatory response to an infectious agent, as well as to promote primary haemostasis (Hamzeh-Cognasse et al., Platelets and infections - complex interactions with bacteria. Front Immunol. 2015).
The impact of LPS in platelets is likely to be of significant relevance. First, LPS directly interacts with platelets, even in the absence of immune cells, and promotes platelet activation and degranulation, P-selectin exposure and in turn activation of leukocytes via PSLG1 binding [14]. Platelets are less sensitive to LPS and therefore may be involved in acute phase of infection.
The involvement of platelets in sepsis is also evidenced in vivo by the fact that as a consequence of LPS injection in mice, platelets rapidly drop out resulting in mild to severe thrombocytopenia. This also occurs in patients with sepsis, in which thrombocytopenia-induced by sepsis is associated with poor prognosis [78. Gonzalez et al., Sepsis and Thrombocytopenia: A Nowadays Problem. Cureus. 2022].
The importance of platelets in sepsis in vivo is now further discussed in section 6.
4- In the section 5.1 authors discuss LPS capability to induce IL1b synthesis by platelets. However, IL1b synthesis has been detected upon platelet storage for more than 24 hours, what should have had a significant impact on platelet state. Thus, please give additional details of the experimental design to the main text.
Shashkin et al. [14] demonstrated that LPS can induce IL-1b synthesis in platelets by real time PCR, flow cytometry and confocal microscopy. They used highly purified platelets, with negligible contamination of monocytes (few per million platelets) obtained by two steps of immunodepletion with a combination of anti-CD14, anti-CD15 and anti-CD45 conjugated to magnetic beads. Anti-glycophorin was used to remove residual erythrocytes. Isolated platelets were stimulated with LPS (10-100 ng/ml) in the presence of autologous serum or recombinant soluble CD14 and LBP. Real time PCR reveals that maximal accumulation of processed IL-1β RNA occurred 3 h after stimulation with LPS. In flow cytometry experiments, WP were stimulated for 0-3 hours and cells were gated with anti CD42 antibody, permeabilized and stained with anti IL-1β. In confocal microscopy experiments, WP with LPS for were stimulated for1-18 hours. Details of the experimental design have now been added in section 5.1.
5- In the section 5.2 the authors discuss that LPS-activated platelets might induce NETosis. Please, discuss whether it is possible that LPS binds to platelet TLR4 and upon platelet interaction with neutrophils platelet TLR4-LPS complex can cause neutrophil TLR4 activation and subsequent NETosis? Finally, please give additional details on the procedures of neutrophil washing as it can significantly affect neutrophil activation.
Clark et al. [48] reported that resting platelets or LPS alone are not able to induce NETosis. However, when both platelets and LPS were added to neutrophils, NETosis occurs. It has been shown that LPS-treated platelets avidly attached to adherent neutrophils under flow conditions (Clark et al., Platelet TLR4 activates neutrophil extracellular traps to ensnare bacteria in septic blood. Nat Med. 2007). This interaction is dependent on platelet TLR4. LPS-induced platelet-neutrophil interactions caused marked neutrophil degranulation, MMP9 release from adherent neutrophils and release of NETs under flow. The authors did not observe NETs when the neutrophils were stimulated with either LPS alone or platelets alone, and this confirms that the procedure of isolation of neutrophils do not activate the cells. NETosis was also observed in vivo: in this case, blood samples from severely septic individuals but not form healthy donors, added to healthy neutrophils and platelets in the presence of DNA dyes promoted the formation of NETs. The requirement of LPS-stimulated platelets in NETosis may be justify by the fact that platelets are less sensitive to LPS than neutrophils, and the authors suggested that platelets may function as a barometer in the blood, becoming activated only under very serious systemic infections to induce neutrophils to entrap bacteria.
Details on the procedures of neutrophil washing are as follow: whole anti coagulated blood was collected, and erythrocytes were removed using dextran sedimentation followed by two rounds of hypotonic lysis. Neutrophils were isolated from the resulting cell suspension using Ficol-Histopaque density centrifugation. Finally, purified neutrophils were suspended in DMEM at a concentration of 107 cells/ml.
In this experimental setting, unstimulated platelets fail to bind to adherent neutrophils and to release NETs and this confirm that the procedure of neutrophil washing maintain neutrophils in a resting state [48].
- In the section 5.5 please add information on the methods of detection of activation of different pathways of LPS-induced platelet activation.
Platelet activation has been analysed using different techniques: in Brown and McIntyre [45], Akt and JNK phosphorylation was detected in ELISA, whereas the majority of studies analyzed phosphorylation of selected substrates in immunoblotting with specific anti-phospho-antibodies [13,16,17,50]. This is now better specified in the text in section 5.5.
7- In the discussion the authors claim that the use of pharmacologic inhibitors and/or transgenic mice can be of help for determination of the LPS role for platelet activation. It would be great to see more specifics on what authors propose.
We thank the reviewer for this suggestion, and we better explained the meaning of the sentence. Actually, the use of different pharmacological inhibitors of selected signaling molecules and the use of transgenic mouse model available will help to elucidating the molecular mechanism of LPS induced-platelet activation, and the discovery of new pathways and new intracellular effectors may lead to the identification of potential targets to modulate thromboinflammation and sepsis.
Minor comment
- Please avoid recurrent description of the LPS structure, LBP and sCD14 present in the section 2.
Redundant parts on LPS structure, LBP and sCD14 have been now removed from section 2 and are inserted in the legend of figure 1 and 2 respectively, as originally planned.